# On Indoor Localization Using WiFi, BLE, UWB, and IMU Technologies

**DOI:** 10.3390/s23208598

**Published:** 2023-10-20

**Authors:** Samuel G. Leitch, Qasim Zeeshan Ahmed, Waqas Bin Abbas, Maryam Hafeez, Pavlos I. Lazaridis, Pradorn Sureephong, Temitope Alade

**Affiliations:** 1Department of Computing and Engineering, University of Huddersfield, Huddersfield HD1 3DH, UK; samuel.leitch@hud.ac.uk (S.G.L.); m.hafeez@hud.ac.uk (M.H.); p.lazaridis@hud.ac.uk (P.I.L.); 2Department of Electrical and Electronic Engineering, University of Bristol, Bristol BS8 1QU, UK; waqas.abbas@bristol.ac.uk; 3College of Arts, Media and Technology, Chiang Mai University, Chiang Mai 50200, Thailand; pradorn.s@cmu.ac.th; 4Department of Computer Science, School of Science and Technology, Nottingham Trent University, Nottingham NG11 8NS, UK; temitope.alade@ntu.ac.uk

**Keywords:** 6G, BLE, data fusion, indoor localization, IMU, PDR, UWB, wi-fi

## Abstract

Indoor localization is a key research area and has been stated as a major goal for Sixth Generation (6G) communications. Indoor localization faces many challenges, such as harsh wireless propagation channels, cluttered and dynamic environments, non-line-of-sight conditions, etc. There are various technologies that can be applied to address these issues. In this paper, four major technologies for implementing an indoor localization system are reviewed: Wireless Fidelity (Wi-Fi), Ultra-Wide Bandwidth Radio (UWB), Bluetooth Low Energy (BLE), and Inertial Measurement Units (IMU). Sections on Data Fusion (DF) and Machine Learning (ML) have been included as well due to their key role in Indoor Positioning Systems (IPS). These technologies have been categorized based on the techniques that they employ and the associated errors in localization. A brief comparison between these technologies is made based on specific performance metrics. Finally, the limitations of these techniques are identified to aid future research.

## 1. Introduction

Indoor localization with an accuracy below ten centimeters has been cited as a key goal for the upcoming Sixth Generation (6G) communications [1]. The reason behind this is its potential to serve as a facilitator for ongoing and forthcoming industrial revolutions. Applications of indoor localization include monitoring patients in care homes [2], asset management and tracking in warehouses [3], navigation of autonomous robots [4], and many more [5]. Localization is the process of determining the location of a target device by taking measurements from certain fixed landmarks. The set of landmarks used by the localization system is referred to as the *map*. The standard technique for achieving localization in an outdoor setting has traditionally been the use of the Global Positioning System (GPS), which uses satellites to determine the position of the receivers [6,7]. Unfortunately, the signals used by GPS cannot penetrate the walls and roofs of buildings, and as such are not suitable for indoor localization [8]. Furthermore, compared to the outdoor scenario, an indoor environment is very challenging owing to exacerbated multipath issues caused by the many reflections and the obstructions that block the direct Line-of-Sight (LoS) link between the transmitter and the receiver. A closely related concept to localization is Simultaneous Localization and Mapping (SLAM), which uses measurements from the target device to generate the map at run-time [9]. This comes with the advantage of not needing details of the map to be programmed into the application. For applications in real-world settings, it is crucial to consider scalability, affordability, setup, and running costs before choosing a localization mechanism for installation.

Several technologies have been investigated for indoor positioning, including Bluetooth Low Energy (BLE) [10], Ultra-Wide Band (UWB) [11], Inertial Measurement Unit (IMU) [12], and Wireless Fidelity (Wi-Fi) [13]. These technologies have a broad penetration in modern homes and workplaces due to their availability in the modern smartphones. This availability has led to a reduction in the unit cost of such devices. For this reason, they have been chosen as the target of this survey. Other technologies available in a modern smartphone or available for low cost have not been considered as the main targets of this survey, which is for a number of reasons. These include range issues, as are present with Long-Range Wide-Area Networks (LoRaWAN), Radio Frequency IDentification (RFID), and 5G, as well as impacts on the lifestyle and behaviour of users, as is the case with acoustic technologies and ultrasound. The target technologies require distinct information for localization, including: (1) proximity, where the location of a user is computed based on the location of the nearest base station; (2) trilateration, where the location of the device is calculated based off of range estimates to multiple available base stations; (3) triangulation, where the target location is calculated based on the angles measured between the target device and multiple available base stations; and (4) fingerprint matching, where a specific feature (such as the Received Signal Strength (RSS)) is mapped for known locations in the target area. New measurements can then be performed and compared with the map to determine the exact device location [5]. Each of these mentioned technologies have their advantages and disadvantages. To maximize advantages while minimizing disadvantages, the Data Fusion (DF) technique is generally employed. DF has been used in a wide range of fields and has proven to be a popular technique in the field of indoor positioning [14,15,16,17].

### 1.1. Related Work

Indoor localization has been an active area of research in the last decade and there are multiple surveys available on this topic. Both Davidson and Piché and Naser et al. focus their papers on the technologies and techniques available on smartphones [18,19]. This particular research direction is very similar to the one taken in the present paper. However, Davidson and Piché’s survey was conducted in 2017, and has become outdated with the advent of technologies such as BLE AoA determination and the inclusion of UWB in modern smartphones. Similarly, Naser et al. did not include UWB into their survey, and did not provide a direct comparison of the performance of different technologies in the literature in terms of their achieved accuracy [18,19]. The authors of [20,21,22] covered a wide range of wireless technologies used for indoor localization, from WiFi to Long-Range Wide-Area Networks (LoRaWAN) and even Frequency Modulation (FM) radio. None of them included inertial methods of indoor localization, which is an extremely popular method due to the low cost of the devices. In [23], the author targeted the field of inertial IPS, focusing entirely on this technology. The paper mentioned how the fusion of an inertial IPS with another IPS based on a different technology is preferable to the inertial IPS alone. This process of DF was the subject of a focused survey in [24], where the authors examined a wide range of technologies and the methods used to fuse them together. They split data fusion into three types: homogeneous, where the system is based on a single technology and either a single measurement type or multiple measurement types; heterogeneous, where more than a single technology type is used with only a single measurement type, e.g., RSS; and hybrid, where multiple technologies and multiple measurements are utilized. This last category contains the fusion of an inertial IPS with other technologies; in [24], the combination with by far the most listed references was found to be the combination of WiFi and inertial. The authors additionally mentioned how Machine Learning (ML) can be applied to the DF problem.

ML has been the centre of several surveys in the literature. The authors of [25] examined a wide range of ML use cases and techniques, providing comparisons between systems with and without ML. They provided a detailed analysis of the current shortcomings of the approaches in the literature as well as potential solutions. The authors of [26] focused their survey of ML on the highly specific area of WiFi RSS fingerprinting, where they covered areas such as the use of ML for crowdsourcing and generating radio maps and other data augmentation techniques along with a detailed overview of the ML techniques used for RSS fingerprinting. The authors of [27,28] provided surveys of ML techniques in the indoor positioning field with a focus on SLAM. The latter focused on visual SLAM, including Visual Odometry (VO), optical flow, loop closure, and many other important areas of SLAM. The former covered the same areas of SLAM while additionally providing direct performance metrics in their analysis of the techniques, similar to our goal in the present paper. Another work focusing on SLAM is [29], which looked at the use of radio technologies for SLAM. The authors provided a detailed overview of the SLAM field, including the datasets available for the development of a radio-based SLAM system without the need for the time-consuming process of gathering one. The authors of [1,5,8] all took more theoretical approaches to their survey work. These reviews covered all aspects of localization, including SLAM and ML; however, none provided a detailed quantitative comparison. Table 1 provides a visual comparison of the topics covered by these related works, including the present paper, sorted in order of year of publication.

### 1.2. Motivation and Contribution

The goal of this paper is to collate and analyze as many papers as possible on a subsection of the indoor localization field to identify which technologies are best suited to practical implementation. The justification for our selection of the reviewed technologies can be found in the next section. The contributions that this work makes to the literature include:An in-depth analysis of technologies used for indoor localization focused on the practicality of the technology for implementation.Key limitations of indoor localization techniques and possible future research directions for indoor localization.

### 1.3. Outline of Paper

The rest of this paper is organized as follows. Section 2 provides a detailed discussion and highlights the different techniques available for an Indoor Positioning System (IPS) along with information on their performance found in the literature. A comparison of the different technologies covered regarding crucial performance metrics is performed at the end of this section. Section 3 focuses on DF, particularly its implementation, benefits, and use cases for indoor localization. Section 4 discusses ML, the techniques that can be used for IPS, and their potential impact on improving localization accuracy. Section 5 discusses limitations of the technologies covered and possible future research directions in this field.

## 2. Technologies

There are a multitude of technologies that can be used for the purposes of localization [14]. Four significant technologies for indoor localization have been selected: (1) Wi-Fi, (2) UWB, (3) BLE, and (4) IMU. These were chosen primarily because of their wide availability in smartphones, which have become ubiquitous in today’s society. Cameras and microphones were discounted due to the disruption they cause to the operation of the smartphone as well as the potential ethical issues around having access to the users voice and image data. Technologies such as LoRaWAN and 5G were not considered due to the relatively low range of the environments considered for this survey. The theory behind this approach was that technologies that are common in smartphones are abundantly available for utilization by other applications, reducing the costs of any potential system.

### 2.1. Methodology

When implementing an IPS, four conditions must be satisfied. First, the implementation cost must be kept low. This includes the cost of the devices and downtime for the area being covered by the system must be kept to a minimum. Using readily available hardware can significantly improve this issue, as can the ability of devices to cover a wide area, which reduces how many devices are needed for the target area. Second, the cost of operating the system must be kept low. This is a more abstract criterion, requiring that the system be as unobtrusive as possible and limit recurring disruptions to the target area. This can be achieved by ensuring that the system consumes as little power as possible, thereby keeping maintenance time and costs to a minimum. Third, the system must be simple to install, requiring little to no surveying or modification of the target area. Fourth, the potential localization error of the system must low in order to achieve the most accurate IPS possible.

For each technology covered above, tables are provided that offer a breakdown of the different techniques used, which papers implement that technology, the range of accuracies reported by those papers, and any advantages or disadvantages of that approach. When the localization accuracy for a paper was not provided, it was calculated by taking the mean of all available accuracy measurements. For instances in which a mean is not calculable and a Cumulative Distribution Function (CDF) was provided, the 50% accuracy measure has been used. For all papers covered in this review, the range is based on the lowest and highest mean accuracies. The citation for the paper that reported the lowest localization error is highlighted in bold font.

The database selected for the initial search was IEEExplore, where a keyword search of “All Metadata”: “TECHNOLOGY” AND (“All Metadata”: “Localization” OR (“All Metadata”: “Indoor” AND (“All Metadata”: “"Positioning” OR “All Metadata”: “Localization”))) was performed, with TECHNOLOGY being replaced by the name of the particular technology being searched for, e.g., BLE or WiFi. These results were then further filtered to return only those journal papers released after 2019. Any papers found using this search that utilized more than a single technology to achieve localization were considered fusion papers. These papers formed the core of the literature considered in this review, and were further augmented by additional papers found using a snowballing technique based on the initially selected papers.

### 2.2. Wi-Fi

IEEE 802.11, commonly referred to as Wi-Fi, is the standard technology for implementing a Wireless Local Area Network (WLAN) in both workplace and home environments. It operates in two main bands, 
2.4
 GHz [10] and 5 GHz [30], utilizing several 20 MHz communications channels within these bands [10]. Owing to the wide availability and low cost of its hardware, Wi-Fi is highly attractive for IPS, especially for factories where low costs are essential. A large portion of the research on WiFi localization centers around the use of RSS for building a fingerprint map. Channel State Information (CSI) has been employed for fingerprinting as well; however, it has two main drawbacks compared to RSS [31]. First, it requires a specific model of Wi-Fi Access Point (AP) to expose the CSI values. Second, it requires special software running on the AP to gain access to the CSI [30]. CSI has many advantages over RSS, including increased stability at a fixed location and the ability to encode the multipath effect into its samples [32]. There are other means of building a fingerprint map for localization [33,34]; however, CSI and RSS fingerprinting are the most common techniques when employing WiFi.

Wi-Fi can be used to measure Round-Trip Time (RTT) through the Fine-Timing Measurement (FTM) protocol added in IEEE 802.11-2016 [35]. Several studies have employed WiFi’s new FTM protocol for localization [36,37]. Systems based on FTM can achieve sub-meter accuracy [36,37]. FTM-based systems are relatively new and have not yet been widely adopted. Only the latest WiFi APs and smartphones are capable of handling FTM requests; more widespread distribution would require the replacement of existing infrastructure. Other disadvantages of FTM include its poor performance in NLoS conditions and decreased ranging accuracy owing to interference from network traffic generated by other WiFi APs operating in the local area [37]. These issues make it less desirable for implementation in an IPS. Table 2 provides an overview of the different techniques available for use in WiFi-based localization, including the range of localization errors that have been reported. As Table 2 shows, the only technique that is capable of achieving localization errors of 10 cm or lower is CSI fingerprinting, which comes with the significant downside of requiring a site survey. The same applies to RSS fingerprinting, which is the next best. RSS ranging performed the poorest in the surveyed papers, possibly because of the unreliability of individual RSS readings.

### 2.3. UWB

UWB radio has several properties that make it highly attractive for use in indoor positioning [56,57,58]. These include: (1) its wide bandwidth, which results in an extremely narrow peak in the time domain, leading to more accurate Time-of-Arrival (ToA) determination [11]; (2) its object penetration characteristics are far superior to other technologies [59]; (3) its low transmission power means that UWB beacons can operate for long periods without maintenance [60]; and (4) it causes very little interference with other wireless signals operating in the same area [60], among many other useful properties [60]. This makes UWB an almost ideal candidate for any IPS. Evidence of this can be found in Figure 1, which shows a constant trend of accelerating research in the field beginning in 2016. Future research on the use of UWB is expected to increase following the release of a new generation of smartphones with UWB capability [61].

UWB technology is capable of incredibly accurate (centimeter level) ranging because of its superior ToF determination [11,62]. As shown in Table 3, it is capable of achieving the sub-10 cm localization errors targeted by 6 G. Most of the research on UWB IPS is based on ranging estimates from Time of Flight (ToF) measurements [63,64]. Despite the improved ability of UWB to penetrate a wide range of objects [60], it remains subject to ranging errors induced by NLoS conditions [64]. To address this, the detection of the NLoS condition [11,65] and mitigation of the resulting ranging errors [65] have become active areas of research. Other studies have focused on channel impulse responses between anchors and receivers [66,67]. The theory behind this approach is to utilize the Multi-Path Components (MPCs) of the signal to enhance localization accuracy [67] in a manner similar to WiFi CSI. This makes UWB Channel Impulse Response (CIR) an ideal candidate for localization in a cluttered environment with many MPCs.

### 2.4. BLE

BLE is a low power consumption subset of Bluetooth developed by the Bluetooth Special Interest Group. The most common use of BLE for localization is through beacons, which are small battery-powered devices that broadcast BLE advertising packets that other devices can locate themselves with. BLE beacons used for indoor localization often implement Apple’s iBeacon protocol [3]. BLE advertisements occur in three 2 MHz wide channels in the 
2.4
 GHz band. This band is the same as that used by WiFi; to avoid interference, the channels are spread out over the band [10]. This leads to unequal signal attenuation of channels owing to the irregular frequency response of smartphone antennas [10] as well as to higher path losses at higher frequencies [72]. This unequal signal attenuation issue needs to be addressed before accurate BLE RSS-based IPS can be designed [72]. Further complicating this issue, standard compliant BLE devices such as those found in smartphones do not include information on the channel index when advertising [10]; therefore, other means, such as the timing of when the advertising packets were received, must be utilized [72].

With regard to localizing smartphones indoors, BLE’s popularity as a technology has seen a rapid increase in recent years, as shown in Figure 1. This rise in popularity is due to all modern smartphones having BLE capabilities and the price of deploying BLE beacons, even in large quantities, being relatively low [3]. BLE-based indoor positioning usually takes the form of RSS-based localization. This can be RSS fingerprinting, RSS ranging with a path–loss model, or using RSS to determine the proximity to specific beacons as mentioned in Table 4. RSS measurements are simple to implement and require no hardware modifications. However, a large number of measurements are required to achieve reliable localization, increasing the time delay of localization estimates [73].

The Bluetooth mesh technology, introduced to provide Wireless Personal Area Networking (WPAN) capability to the Bluetooth stack, has received attention in recent years as well [84]. The authors of [84] suggested that the technology is poorly suited to applications that cover a wide area due to the heavy impact on localization delay caused by introducing new nodes to the mesh network, reporting a localization error of 
4m
.

The release of the Bluetooth 5 standard in 2016 introduced the new feature of extended advertisements which can be used to measure RSS values on all 40 Bluetooth channels [85]. This channel diversity can be exploited to improve the localization accuracy of BLE systems by using a variety of RSS measurements [85]. Bluetooth 
5.1
, released in 2019, added the Constant Tone Extension (CTE) protocol. This opens up new possibilities for BLE-based localization by providing a mechanism for estimating the Angle of Arrival (AoA) of a signal. The phase of the received BLE signal at each antenna is calculated through the In-Phase/Quadrature (I/Q) samples generated by the Bluetooth demodulator [81].

The inclusion of the CTE protocol in the Bluetooth specification enables the use of hybrid AoA/ranging techniques to provide location estimates using only a single anchor. The authors of [86] used a Kalman Filter (KF) to combine a range estimate with an AoA estimate as well as a PDR. However, despite the great promise of BLE AoA, only a small number of papers have been published on this subject [81,82,83,86], as the technology is very new, with the standard defining the technology only being released in 2019.

Overall, localization errors for BLE techniques leave much to be desired. As can be seen in Table 4, only the RSS fingerprinting technique seems capable of 10 cm localization errors at this moment. However, BLE technology is a good choice for an IPS owing to its low hardware cost and power efficiency, which can significantly reduce the costs of deployment and maintenance compared to other wireless technologies.

### 2.5. Inertial Measurement Unit (IMU)

IMUs, often referred to as motion sensors, are Micro-ElectroMechanical Systems (MEMS) that are commonly found in smartphones [15]. As such, they are ubiquitous and cheap, with easy access to data through any smartphone Application Programming Interface (API) and very low power consumption. The acceleration data gathered by the IMU can be used for PDR. Owing to the fact that the location estimate of PDRs has a tendency to drift over time, IPS based solely on inertial data are relatively rare in the literature. Instead, when an IMU is used for indoor localization, it is common to pair it with one or two other technologies in a process called DF, which is discussed in more detail in the next section.

The use of IMU data has proven to be a popular aspect of indoor localization, particularly in smartphones [35]. This is because its availability [15] and independence from electromagnetic signals allow it to be reliably used in complex environments where LoS to a beacon cannot be guaranteed [87]. Furthermore, an IMU-based IPS requires no additional infrastructure for its operation, thereby limiting the disruption caused to any potential location in which the finished IPS would be deployed. Localization accuracies for PDR, by far the most prevalent technique for indoor localization, often fall short of the 10 cm goal due to the error drift mentioned previously.

### 2.6. Comparison

Figure 2 shows a radar graph which provides a visual summary of the performance of the various techniques covered above with respect to the performance criteria outlined in Section 2.1. Each subfigure covers the techniques implemented on each technology, with each being ranked as either poor, medium, or good on each of the criteria. As a companion to Figure 2, Table 5 compares the technologies on other categories which are worth considering during selection, as suggested in [88]. For indoor localization, the goal is to achieve an average estimation error of less than 10 cm. The only technologies capable of achieving this level of accuracy on their own are WiFi and UWB, as can be seen in Table 2 and Table 3. UWB hardware is relatively niche, and would need to be installed into the target building for operation. Even with UWB hardware now being affordable [89], this would drive up the costs of installing a UWB system [90], resulting in its poor ranking in Figure 2b. In contrast, WiFi hardware is ubiquitous and there is a high chance of it already being available at the target location; this is reflected in Figure 2a. As Figure 2c demonstrates, BLE fits the first two criterion for implementation and operational costs perfectly, with the sole exception of the fingerprinting technique. For both BLE and WiFi, fingerprinting performs poorly on three of the four criteria, although it does tend to provide among the best localization accuracies for the given technologies. Unfortunately, Table 4 shows the difficulty of achieving the level of accuracy demanded by 6G. The PDR technique is possibly the easiest to implement and cheapest to install, as represented in Figure 2d. This is because it only relies on the target device for implementation. However, the phenomenon of error drift, where the localization error increases with time, causes significant issues. Combining two or more technologies together has proven to be very effective in reducing the localization error while maintaining low power consumption. The next section describes the DF process in more detail.

### 2.7. Other Technologies

The technologies mentioned in this section are not the only ones that can be used for IPS. A wide range of other technologies are often utilized, such as RFID, ZigBee, Visible Light Communication (VLC), and acoustic. While these were not chosen as the main subjects of this survey for various reasons, they are important and prevalent for IPS.

#### 2.7.1. Radio Frequency IDentification (RFID)

Ultra-High-Frequency (UHF) RFID is the main type of RFID employed for IPS due to its significantly increased range over the other forms of RFID [91]. UHF RFID is capable of centimeter-level localization accuracy; however, to achieve this, it needs an average beacon deployment density of 1 beacon every square metre [91]. This is impractical and a system designed for it would be intrusive. Furthermore, this density requirement would increase the cost of the system when deployed in large areas, even with inexpensive RFID beacons.

#### 2.7.2. ZigBee

ZigBee is a communication protocol built on the IEEE 802.15.4 standard [22]. As with BLE and WiFi, ZigBee uses RSS measurements for localization, and suffers from poor localization errors as a result [92]. While ZigBee’s hardware is inexpensive, its lack of availability on user devices means that it loses out to WiFi for use in IPS [21].

#### 2.7.3. Visible Light Communication (VLC)

VLC-based localization can be achieved using a variety of techniques, including proximity and triangulation. It can be realised using practically any Light Emitting Diode (LED) light source, including lights that already exist in indoor locations [93]. Unfortunately, VLC is highly susceptible to NLoS conditions [21], and its implementation for personal use requires access to either a camera on the user device, potentially opening up ethical issues around the users’ private image data, or a rotating light sensor placed on the user in an unobtrusive position.

The authors of [93] suggested a novel means of realizing VLC in through so-called passive approach that allows the light sensors to be independent of the target device. This approach has a lower accuracy bound on the order of tens of centimeters [93].

#### 2.7.4. Acoustic

Acoustic-based localization utilizes sound waves modulated with a time stamp to perform ToA-based trilateration. Acoustic-based localization is capable of centimeter-level accuracy, and is extremely suited to smartphone platforms [94]. Unfortunately, the cost of the hardware used for the anchors is much higher than that of other technologies [94], and the need to use smartphone microphones means that the audio signal must be in the audible range [94]. Another significant factor to consider is the need for constant access to a microphone which is carried by the user. Thus, measures must be put in place to protect users’ private audio data in order to ensure trust in the system.

## 3. Data Fusion

In this paper, DF refers to the use of multiple technologies and/or techniques within a single IPS in order to provide an overall improvement in accuracy, reliability, and coverage. This differs from the classical definition in that, to a certain extent, most IPS instances are already based on fusion, for instance, trilateration, where range measurements from at least three different beacons enable localization in two dimensions, and triangulation, which relies on AoA measurements from at least two different beacons in two dimensions. The only exceptions are CSI-based hybrid localization systems, which enable localization from a single beacon using a single measurement [48,52,53,54]. Theoretically, fingerprinting techniques are capable of localizing using a single beacon, and would fall into this latter group; however, most practical applications use multiple beacons [10] or fuse them with other technologies [86]. This is because of the ability of DF to maximize the advantages of technologies while minimizing their negatives [95], such as reducing an IPS sensitivity to RSS fluctuations and providing landmarks to reduce the error drift in PDR.

### 3.1. Kalman Filter (KF)

Despite the process being decades old, KF or variants such as Extended KF (EKF) and Unscented KF (UKF) [96], remains the predominant method of implementing DF. A KF is an “optimum state estimator” [96] that propagates an estimate of the system state from the previous time step forward in time and then uses measurements to ensure the correctness of this estimation [97]. For a given a system described by the state vector 
x
, which evolves in a manner described by the state transition matrix 
H
, an estimate of the current system state at time *k* can be made as follows:
(1)
$${\hat{x}}_{k|k-1}=H\cdot {\hat{x}}_{k-1}+v\left(k\right),$$

where 
v(k)
 is the system noise and 
${\hat{x}}_{k-1}$
 is the previous system state estimation. When a measurement of this system is made, it represents an approximation of the true state of the system. This can be represented mathematically as

(2)
$$z_{k}=M\cdot x_{k}+w\left(k\right),$$

with 
M
 being the observation matrix mapping the state vector 
x
 to the measurement vector 
z
 and 
w(k)
 being the measurement noise. This measurement vector can then be used to update the state estimate

(3)
$${\hat{x}}_{k}={\hat{x}}_{k|k-1}+K_{k}\left(z_{k}-M\cdot {\hat{x}}_{k|k-1}\right),$$

where 
$K_{k}$
 is the Kalman gain [98]. The Kalman gain can be understood as a mathematical measure of how much the initial state estimate can be trusted. Data fusion can be achieved in Kalman filtering through a number of means, such as concatenating the individual measurements into a single measurement vector [99]. Alternatively, separate KFs can be applied to the measurements and then their individual outputs can be combined. Another solution is introduce an intermediary step to extract information from the measurements, with the KF being applied to these new features [100].

EKF and UKF are modifications of KF that were developed to address the problems that arise when using KF with nonlinear systems. EKF works by linearizing the system equations by calculating the Jacobian matrix of the system equation and then substituting a linearized approximation [97]. The regular KF process is then applied to the linearized state equations [96]. UKF works by generating a set of points that perfectly capture the statistical information of the underlying system and then propagating these points through the nonlinear system equations [97]. UKF provides several advantages over EKF, such as working with discontinuous functions and eliminating the need for calculating Jacobian matrices while offering comparable performance in terms of the number of required computations [97].

### 3.2. Inertial Measurement Unit (IMU) Fusion

As mentioned in the previous section, IMUs are extensively used for localization in smartphones owing to their low cost, wide availability, and independence from electromagnetic interference. The array of wireless communications technologies they present combined with their ubiquitous presence on smartphones means that DF of IMU data and other localization techniques is inevitable. In fact, IMUs are rarely used alone because of the inherent disadvantages of PDR and the low accuracy of smartphone IMUs. Furthermore, IMUs can provide reliable positioning estimates when RF technologies cannot, such as in NLoS conditions or when the device is out of range of the beacons [35]. This makes the resultant IPS much more resilient in the harsh propagation conditions presented by indoor settings. A summary of localization accuracy results from papers that have fused PDR with other techniques can be found in Table 6.

### 3.3. Crowdsourcing

One of the more popular means of localization through RSS measurements for both WiFi and BLE is building and using a fingerprint map, as mentioned in Table 2 and Table 4. This is because it does not require any hardware or software modifications to the underlying infrastructure or smartphone and can operate with only a single beacon. The major problem with fingerprinting is that thousands of samples must be gathered at each reference point. Slight changes in the environment can affect how the signal propagates through it, rendering the original map inaccurate and useless. This necessitates the repetition of the site survey for a real system, driving up the cost of maintaining an IPS based on fingerprinting [106].

To mitigate these issues with the fingerprinting process, several papers have been published which use a process called crowdsourcing. Crowdsourcing aims to avoid the need to perform a site survey each time the map needs to be updated. It does this by using data gathered by users in the area to automatically update the map. This process requires that the user be accurately localized in the first place, otherwise the accuracy of the resulting localization system is severely affected [105]. Another significant issue with crowdsourcing is the security risks stemming from potential leaking of personalized user data. Comparing the localization accuracy for crowdsourcing in Table 6 to the localization accuracies of the RSS fingerprinting technique in Table 2 and Table 4 shows no difference. This is great news for this, technique as it could completely eliminate the need for site-surveys.

### 3.4. Hybrid, Heterogeneous, and Homogeneous Fusion

Hybrid, heterogeneous, and homogeneous fusion are terms defined by the authors of [24] in their survey of fusion-based IPS. The definitions are as follows:Hybrid: the combination of disparate measurements from different technologies to unlock the full potential of the measurements. While this is primarily realized in the literature through the combination of IMU measurements with other technologies, as outlined in Section 3.2, this is not the only hybrid fusion of technologies available.Heterogeneous: the combination of a single measurement type gathered from different technologies. An example of this could be the fusion of RSS measurements from WiFi, BLE, and RFID to improve fingerprinting localization [109].Homogeneous: the combination of different measurement types from the same technology, e.g., the combination of WiFi RSS and FTM measurements [110].

## 4. Artificial Intelligence (AI)

The application of AI to indoor localization can provide several benefits to the problem domain. This is largely due to its ability to learn potentially abstract patterns that are present in the training data. There are many ways that ML can be applied to the localization task. One popular application is the use of the K-Nearest Neighbors (KNN) algorithm for fingerprint matching [10,26]. This algorithm compares the received signal features to every available reference point and finds the *k* reference points which are the closest match for the received signal. The final position of the target is then determined to be the weighted average of the positions of the selected *k* reference points [112]. The Naïve Bayes Classifier (NBC) is another classifier. It uses the Bayesian theorem to determine the likelihood of a signal belonging to a certain class based on the combined probability of it having certain features. It has been used for indoor localization to detect LoS conditions in UWB [11]. Other classifiers include the Decision Tree (DT) algorithm, which creates a branching network of decision boundaries that eventually converge on the class, and Support Vector Machine (SVM), which determines the plane which optimally separates the classes. Regression based on the SVM algorithm was utilized in [113] to automatically update the radio map in a crowdsourcing-based fingerprinting scheme. A common neural network-based alternative to SVM for fingerprinting is the use of Extreme Learning Machine (ELM). This neural network architecture typically consists of a single hidden layer, and its unique training process means that it can make accurate predictions based on new samples with just a single pass through the training dataset [114,115,116]. The Long Short-Term Memory (LSTM) model is often used to perform PDR-based localization, where its capability to infer outcomes based on time series data has proven invaluable in improving the localization error [39].

### 4.1. Deep Neural Networks (DNNs)

DNNs are a class of ML model that are made up of several layers. DNNs cover a wide range of model architectures, including Artificial Neural Networks (ANNs) and Convolutional Neural Networks (CNNs). In an ANN, each layer is made up of one or more neurons with output value that are calculated by multiplying the output value of each neuron in the previous layer with its associated weight:
(4)
$$y_{n}=f\left(w_{n}^{T}h_{n}+b_{n}\right),n=1,2,\ldots ,N,$$

where 
$w_{n}$
 is the weight vector of the *n*^th^ neuron, 
$h_{n}$
 is the input vector of the *n*^th^ neuron, 
$b_{n}$
 is the activation bias of the *n*^th^ neuron, 
f(·)
 is the activation function applied to the neuron’s output [117] and *N* is the total number of neurons in the layer. The output of each neuron is then propagated forward to the next layer, as demonstrated in Figure 3. These layers of neurons, where each neuron in one layer is connected to every neuron in the next, are known as fully connected layers. Due to their interconnected nature, models made up of these layers scale very poorly when the number of neurons in a layer is increased. DNNs have been applied to a wide range of tasks in indoor localization [70,118,119]. One example application of an ANN to indoor localization is [120], which detects LoS blockages on BLE technology and corrects the corresponding RSS dips.

#### 4.1.1. Convolutional Neural Networks (CNNs)

A CNN is a DNN that contains one or more convolutional layers which perform the function

(5)
y(t)=(x∗w)(t)=∫x(a)w(t−a)da,

where *w* is known as the kernel. The kernel is learned by the model in order to perform the assigned task [117]. CNNs are made up of one or more layers which implement this convolution function, and are capable of handling two-dimensional inputs such as images. Use of CNNs in the literature is often focused on WiFi CSI, which can be processed into images [32,50,121].

#### 4.1.2. Recurrent Neural Networks (RNNs)

RNNs are another subclass of DNNs; they incorporate information gleaned from a previous iteration in order to inform the decision at the current time step. These networks are excellent at predicting time series data [122]; however, during the backpropagation process the loss gradient has a tendency to dwindle until it has little to no effect on the weights of the network. This is known as the vanishing gradients problem, and is a significant problem affecting RNN architectures [122].

One solution to this problem which is commonly encountered in the literature is to use an LSTM network. These consist of a number of LSTM cells made up of a forget gate, an update gate, and an output gate [122,123]. The combined function of these gates allows a cell to choose its inputs as well as to select the information that is propagated at each time step [39]. LSTM networks are particularly well suited for applications in PDR, where their ability to analyze time series data has proven to be invaluable [39,122,123].

### 4.2. Discussion

One main disadvantage of ML techniques is their reliance on the availability of relevant data. A large amount of data is required in order for supervised ML techniques to learn the desired model [124]. Collecting the required datasets can be a time consuming process. A major limiting factor on research into ML for IPS is the fact that there are not enough datasets available for public use. To address this issue, research is being conducted on the use of Generative Adversarial Networks (GANs) to generate synthetic training data in order to augment real data gathered experimentally [50]. A GAN works by training two separate models in tandem with each other, which are called the generator and the discriminator. The generator’s task is to take random noise as input and generate an output that matches the training samples as closely as possible. The discriminator’s task is to determine whether its input is a generated input or one of the original training samples [117]. When training is complete, the discriminator can be discarded, leaving a model capable of generating data which closely match the real-world data.

Another issue with training an ML model is that large models require a significant amount of time to train, especially when dealing with complex models and large datasets. This is because the error generated by each sample in the dataset must be differentiated and backpropagated through each layer in the model. This scales poorly for large models with many neurons. This process can be accelerated through the use of more specialized computing resources, such as Graphical Processing Units (GPUs) or the more modern Tensor Processing Units (TPUs) [125].

In order to provide a comparison of various ML models, three different classifiers were trained to perform the task of LoS determination on a synthetic BLE AoA dataset. The dataset was generated using ray tracing to trace the path taken by signals from a BLE transmitter to a 4 × 4 array of BLE receivers. As per the BLE 5.3 specification, the receiver generated an I/Q sample pair at each sampling slot, with eight pairs being collected from the reference antenna before sampling the rest of the array [126]. The classifiers were trained on one I/Q sample per antenna and four I/Q samples per antenna, respectively, to determine the effect on the size of the final model and the size of the data vector needed for inference. The chosen classifiers were SVM, KNN, and DT algorithms. The classifiers were evaluated on precision, recall, and accuracy as well as on the final size of the model. The results of this analysis can be seen in Figure 4. Using one sample per antenna instead of four results in a 
66.2%
 reduction in the size of the input data vector while having negligible effects on the performance of the final model, as demonstrated in Figure 4. Furthermore, there are significant benefits in terms of the size of the final models. KNN, while performing the best in terms of precision, recall, and accuracy on both of the tests, performs far worse in terms of disk size. This is because KNN works by comparing the input vector to every single instance in the dataset in order to determine which samples it is closest to. This has significant implications for its potential use in TinyML operating on severely memory-limited devices.

To further demonstrate that ML can provide significant benefits in the indoor localization field, Table 7 lists a number of papers which have reported the accuracy of their IPS both before and after the application of ML, showing the calculated percentage improvement. From these results, it can be observed that ML provides a significant improvement in localization capability when applied to IPS.

As mentioned in Section 3, there has been a continual increase in the use of ML for DF owing to its ability to accurately learn complex relationships between data [103,128]. Similar to KF, DF can be implemented through ML by either measurement fusion, by combining the measurement vectors of different sensors for use as input to the model, or by state vector fusion, where features are extracted from the individual sensor modalities before being passed to the model for inference [103]. Decision fusion is the third potential method for DF using ML. It relies on the generation of individual location estimates which can then be combined to form a more accurate final decision. ML can be applied to the models used to generate the initial estimates, a final classifier that averages or weights the individual decisions, or both.

## 5. Future Research Directions and Challenges

Indoor localization is a relatively mature area of research [70], opening up the field to applications in a wide number of areas such as care homes, malls, and disaster relief as well as for asset tracking in factory environments. To help identify potential future trends in IPS, a survey of the papers published in the field was performed using Scopus. Figure 1 shows the results obtained from Scopus regarding the number of documents published each year for the reviewed technologies. It can be observed that most of the research since 2020 has been focused on visual localization techniques such as Visual Inertial Odometry (VIO) or on UWB. The clear upward trend in the number of papers published per year shows the growing popularity of the indoor localization field, with UWB and camera-based localization showing almost exponential growth in research interest.

### 5.1. Future Research Directions

With the advent of BLE AoA technology, a new avenue of research has opened up for triangulation using BLE. This has already gathered significant interest in the literature [81,130,131] and is showing very interesting results in localization accuracy. Table 4 demonstrates the improvements with triangulation compared to traditional RSS-based localization. The ability to access angle information from BLE devices opens up the opportunity for localization using a single anchor, which can significantly reduce IPS installation costs thanks to the possibility of fusing several localization estimates from a range of measurements and individual location estimates. Researchers are beginning to investigate the possibility of AoA determination on UWB devices as wll [132,133].

In [70], the authors used a UAV with a UWB sensor onboard to act as a mobile AP gathering signal data from transmitters inside an arbitrary location. Where UWB sensors are already available, this solution allows for localization in areas where there is no indoor localization system already in place and no means of installing one. Such a solution is ideal in disaster relief situations such as locating victims of earthquakes or other natural disasters which may lead to people being trapped in rubble.

As discussed in Section 4, ML is proving to be a vital future component of IPS. Unfortunately, the need to transmit data from the devices collecting the data at the edge to a central server capable of performing inference on the collected data opens up a number of security and privacy issues. One method which is being developed to address this particular challenge is Federated Learning (FL), in which each edge device learns a model in parallel and then transmits the parameters to a central server which aggregates them and then redistributes the aggregated model back to the edge devices [134,135,136]. This method can be used to make fingerprinting-based IPS more robust against changes in the environment and location [135]

As mentioned in Section 2.7, a new form of passive VLC localization has been proposed in [93] which could allow for device-free localization at centimeter accuracy. This would be exceptional for future IPS, requiring only the installation of sensors in the ceilings of target locations. Other forms of device-free localization are being researched utilizing technologies such as WiFi CSI fingerprinting [49] and UWB [66]. Improvements to fingerprint-based device-free localization is being investigated with the development of meta-learning [137,138], where a separate ML model learns to output model weights for a given novel environment. The application of this technique to other areas of indoor localization could drastically improve the adaptability of systems to new environments.

### 5.2. Challenges

Despite the opportunities that indoor localization provides, a number of open problems need to be resolved before it can be implemented to support future applications.

Survey Delays: As shown in Table 2 and Table 4, the fingerprinting technique is capable of achieving a very low localization error. Unfortunately, the need for a protracted period of site surveying to build the fingerprint map means that the operating environment almost certainly needs to halt operations for health and safety reasons. This problem is exacerbated by the fact that the propagation of WiFi and BLE is heavily influenced by objects in the environment. Factors such as objects or even people moving in the environment can drastically affect the RSS or CSI signature of a point in space [38], invalidating the fingerprint map and significantly reducing localization accuracy. Unfortunately, the solution of crowdsourcing for this problem introduces a new issue that is yet to be properly addressed, namely, the requirement that an accurate estimate of the user’s location already be in place before their contribution to the global map can be useful.Localization Delays: RSS measurements for path–loss modeling are known to deviate significantly even when the subject is standing still [10]. Our literature survey revealed several attempts to address this issue, including applying a KF to RSS readings before localization [74] or simply taking several samples and finding their average [38]. The latter approach can introduce severe time delays into localization estimation, resulting in reduced efficacy when localizing moving objects. On the other hand, the KF-based approach is computationally intensive owing to the need to calculate an inverse matrix. KF has poor performance when tracking a measurement value that has the potential to change, i.e., RSS values when a subject is moving. Research into methods for smoothing RSS values without introducing significant time delays while retaining the ability to tracking moving values could be of significant value in the context of indoor localization.NLoS Errors: One of the primary problems for systems based on ToA and AoA is their reliance on LoS for accurate estimates. One method used in [79] to deal with this is to use the floorplan of the area to be localized to calculate the placement of beacons to ensure that every point of the floorplan is covered by the desired amount of beacons, thereby circumventing the NLoS issue. While this approach has many advantages, it can lead to a very large number of beacons needing to be installed, which could raise the capital expenditure of the system, perhaps very significantly in the case of UWB. Other methods include detection of the NLoS condition, which can then be accounted for in the range estimation process [11]. Further research into the detection and mitigation of NLoS issues is required if techniques such as trilateration and triangulation are to be successfully deployed.Limited Access: With the improved penetration, narrow peak, and robustness to electronic interference provided by UWB technology, there has been a large amount of research into its application [60,69]. The accuracy of systems based on this technology is able to achieve a decimeter-scale localization error; however, such systems cannot provide concurrent access to many users simultaneously [61]. This issue needs to be resolved if centimeter-level localization is to be reached using UWB in crowded areas.Lack of Datasets: One promising area of research has proven to be the application of ML to various aspects of indoor localization. As mentioned in Section 4, one of the major issues facing research on ML is the lack of available datasets. Most experiments are performed in laboratory or office environments, which are only broadly comparable to other indoor environments. Compounding this issue is the fact that very few researchers have released their datasets for public use. If research into the use of ML for indoor localization is to progress, more datasets covering a broader range of environments need to be released. The datasets necessary for training a model for indoor localization require several modes of data, for example, IMU, BLE, UWB, etc., in order to be as universal as possible. For example, a dataset used for the development of an indoor localization system for smartphones could incorporate inertial sensors for PDR as well as multiple forms of wireless communications, including UWB for the most recent smartphones [61]. One alternative solution for this issue could be the application of research into transfer learning or semi-supervised learning techniques. This can allow existing datasets to be extended without requiring extra data to be gathered and stored. Unsupervised techniques could be a potential research direction as well, as they can avoid the need for a lengthy and potentially inaccurate process of labelling the training data, which is one of the main barriers to the creation of new datasets. GANs can be used to address this issue. As previously mentioned, GANs can learn how to imitate training samples. Further research could be conducted on the applications of GANs to supplement the gathered data.Reliance on Devices: Every technique mentioned in this survey requires the user being localized to carry a device with them, such as a smartphone, tag, etc. However, in certain circumstances this may not always be possible. For example, in high-security manufacturing contexts employees may not be allowed to use their phones in sensitive areas. In such cases, localization using these techniques is impossible. Research into device-free localization, such as that performed by [49,66], would go a long way towards mitigating this issue.

## 6. Conclusions

In this paper, we have examined the problem of implementing an accurate IPS. The indoor environment provides many unique challenges, such as exaggerated multipath effects, interference, a highly dynamic environment, and increased sensitivity to disruptions. WiFi, UWB, BLE, and IMU technologies have all been considered as the basis of IPS because of their wide availability in smartphones. For each of these technologies, we have surveyed the papers released since 2019, identified the techniques used within them, and created tables to categorize the publications and display the range of localization errors encountered in each case. We have discussed both DF and ML in depth because of their importance in research seeking to reduce the localization error of IPS. This review provides perspectives on the limitations faced by research in the field of IPS and the possibilities for potential future research directions.

## Figures and Tables

**Figure 1 sensors-23-08598-f001:**
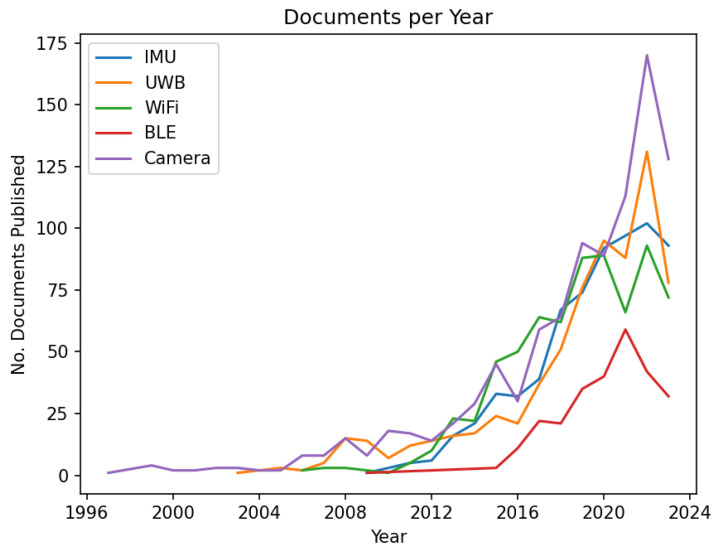
Documents per year for different indoor positioning technologies. Results obtained from Scopus.

**Figure 2 sensors-23-08598-f002:**
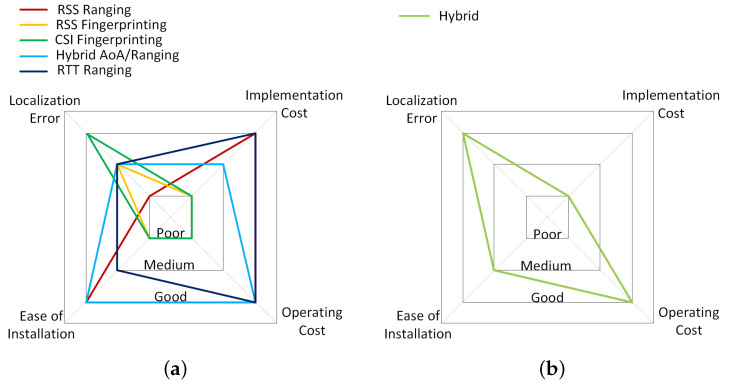
Visual comparison of the performance of different techniques within each technology covered: (**a**) WiFi, (**b**) UWB, (**c**) BLE, (**d**) IMU.

**Figure 3 sensors-23-08598-f003:**
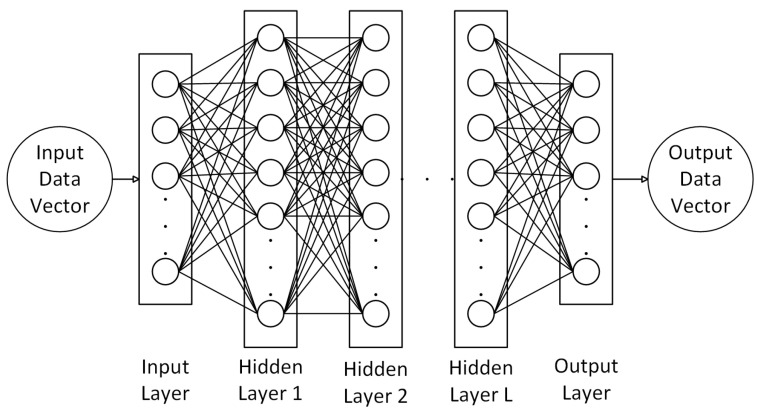
The general structure of an ANN.

**Figure 4 sensors-23-08598-f004:**
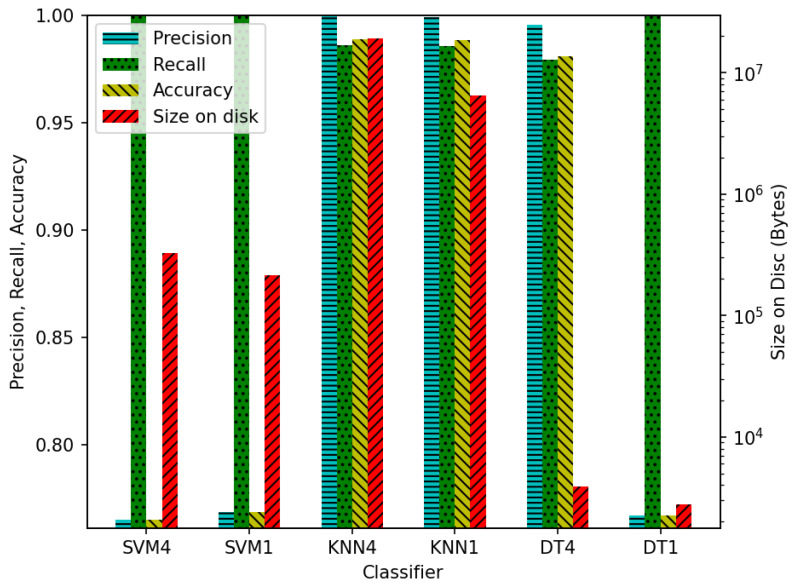
Comparison of different ML models trained to perform LoS determination on a synthetic BLE AoA dataset.

**Table 1 sensors-23-08598-t001:** Table comparing the topics covered by related works and this paper.

		Topics															
Paper	Year	Inertial	Smartphones	WiFi	BLE	UWB	SLAM	V-SLAM	LoRaWAN	DF	ML	RFID	ZigBee	VLC	Acoustic	Ultrasound	5G
[23]	2013	✓	✓	✕	✕	✕	✕	✕	✕	✕	✕	✕	✕	✕	✕	✕	✕
[18]	2017	✓	✓	✓	✓	✕	✕	✕	✕	✓	✓	✕	✕	✕	✕	✕	✕
[1]	2017	✓	✕	✓	✕	✓	✓	✕	✕	✓	✓	✕	✕	✕	✕	✕	✕
[8]	2018	✕	✕	✓	✓	✕	✕	✕	✓	✕	✕	✓	✓	✓	✕	✕	✕
[5]	2018	✕	✕	✓	✕	✕	✓	✕	✕	✓	✕	✕	✕	✕	✕	✕	✓
[21]	2019	✕	✕	✓	✓	✓	✕	✕	✓	✕	✕	✓	✓	✓	✓	✓	✕
[24]	2020	✓	✓	✓	✓	✓	✕	✕	✕	✓	✓	✓	✓	✓	✕	✕	✓
[20]	2020	✕	✕	✓	✓	✓	✕	✕	✓	✓	✕	✓	✓	✓	✕	✓	✓
[27]	2020	✓	✕	✕	✕	✕	✓	✓	✕	✕	✓	✕	✕	✕	✕	✕	✕
[25]	2020	✕	✕	✓	✓	✓	✕	✕	✕	✓	✓	✓	✓	✕	✕	✕	✕
[22]	2021	✕	✕	✓	✓	✓	✕	✕	✓	✕	✕	✓	✓	✓	✓	✓	✓
[26]	2021	✕	✕	✓	✕	✕	✕	✕	✕	✕	✓	✕	✕	✕	✕	✕	✕
[29]	2023	✕	✕	✓	✕	✓	✓	✕	✕	✕	✓	✓	✕	✕	✕	✕	✕
[28]	2023	✕	✕	✕	✕	✕	✕	✓	✕	✕	✓	✕	✕	✕	✕	✕	✕
[19]	2023	✓	✓	✓	✓	✕	✕	✕	✕	✓	✓	✓	✕	✕	✓	✕	✕
This	2023	✓	✓	✓	✓	✓	✕	✕	✕	✓	✓	✓	✓	✓	✓	✕	✕

**Table 2 sensors-23-08598-t002:** Summary of techniques for WiFi-based IPS.

Technique	Advantages	Disadvantages	Accuracy (m)
RSS Ranging [38]	- Low cost of	- RSS values fluctuate	2.397
	implementation	when stationary	
		- No hardware	
		modifications required	
RSS	- Low cost of	- Extensive site survey	0.169–7.6
Fingerprinting	implementation	required before	
[39,40,41,42,43,44,45,46,47]	- No hardware	implementation	
	modifications		
	required		
CSI	- Capable of low	- Requires specific	0.09–2.087
Fingerprinting	localization errors	hardware and	
[32,48,49,50],		firmware to access CSI	
		- Extensive site survey	
		required before	
		implementation	
Hybrid AoA/	- Capable of	- Requires specific	0.75–1.5
Ranging	localization with	hardware	
[51,52,53,54],	a single anchor		
RTT Ranging	- Supported on	- Poor performance	0.5-2.10
[36,37,38,55],	commodity devices	in NLoS	
		- Disrupts regular	
		network traffic	

**Table 3 sensors-23-08598-t003:** Summary of techniques for UWB-based IPS.

Technique	Advantages	Disadvantages	Accuracy
			(m)
RTT Ranging	- Extremely accurate	- Requires specific	0.03–0.28
[56,59,63,68,69,70],	estimates	hardware	
	- Low power	- Susceptible to	
	consumption	NLoS errors	
	- Does not interfere	- Only available in	
	with WLAN	modern smartphones	
Hybrid AoA/	- Utilizes multipath	- Only available in	0.16–0.3
Ranging	components	modern smartphones	
[66,67,71]	- Interference resistance		

**Table 4 sensors-23-08598-t004:** Summary of techniques for BLE-based IPS.

Technique	Advantages	Disadvantages	Accuracy
			(m)
RSS ranging	- Low cost of	- Requires multiple	0.59–4.92
[2,74,75]	implementation	readings or filtering	
	- Requires no	to be accurate	
	hardware modifications		
RSS	- Low cost of	- Requires extensive	0.1–3.73
Fingerprinting	implementation	site survey	
[76,77,78]	- Requires no	before implementation	
	hardware modifications		
RSS Proximity	- Requires no	- Requires multiple	-
[73,79,80]	hardware modifications	readings or filtering	
		to be accurate	
		- Provides location in	
		terms of the nearest	
		beacon	
Triangulation	- Can localize	- Susceptible to	0.7
[81,82,83]	with two beacons	NLoS errors	
	- Supported by newest	- Not supported by	
	generation of	smartphones	
	microcontrollers		

**Table 5 sensors-23-08598-t005:** Table comparing the relative merits of the different technologies covered in this paper.

	Technology	WiFi	BLE	UWB	IMU
Metric					
Accuracy (m)		0.09–7.60	0.10–4.92	0.03–0.30	0.55–5.64
Robustness		Med	Poor	Good	Good
Scalability		Good	Good	Med	Excellent
Security		Med	Med	Med	Excellent
Complexity		Med	Good	Med	Med

**Table 6 sensors-23-08598-t006:** Summary of techniques used for data fusion-based IPS.

Technique	Advantages	Disadvantages	Accuracy (m)
IMU Fusion	- Robustness to NLoS	- Requires	0.18–2.674
[3,15,86,87,95,101,102,103]	conditions	computation	
	- Provides an estimate	resources	
	of user position		
	- Can reset error drift		
Crowdsourcing	- Does not require	- Requires accurate	0.1–4.3
[102,104,105,106,107]	initial site survey	position estimates	
	- Can update maps	for map updates to	
	dynamically	be reliable	
Hybrid Fusion	- Counteracts the	- Requires	0.39
[14,108]	negative aspects	computation	
	of involved technologies	resources	
	- System becomes robust	- Requires	
	against failures	communication	
		resources	
Heterogeneous Fusion	- Makes system more	- Requires	0.6
[109]	robust to device failure	computation	
	- Can counteract	resources	
	the negative aspects	- Requires	
	of involved technologies	communication	
		resources	
Homogeneous Fusion	- Makes system more	- Requires	0.03–0.6
[60,65,110,111]	robust to device failure	computation	
	- Makes system more	resources	
	robust to negative	- Requires	
	environment effects	communication	
		resources	

**Table 7 sensors-23-08598-t007:** Results demonstrating the efficacy of ML for improving IPS accuracy.

Paper	Localization Technique	ML Technique	Reported Accuracy (m)		Percent Improvement
			Without ML	With ML	
[32]	WiFi CSI Fingerprinting	CNN	4.8	2.39	50.2
[116]	WiFi RSS Fingerprinting	ELM	10.64	7.58	28.76
[122]	WiFi RSS Fingerprinting	LSTM	1.47	0.75	48.98
[127]	BLE RSS Fingerprinting	DNN	16.6	5.5	66.87
[127]	BLE RSS Fingerprinting	DRL	16.6	8.6	48.19
[15]	BLE + PDR Fusion	RNN	1.45	1.235	14.83
[128]	BLE + PDR Fusion	DNN	111.36	45.41	59.22
[129]	PDR	LSTM	47.96	2.21	95.39

## Data Availability

Not applicable.

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
