# Peer review of "On Indoor Localization Using WiFi, BLE, UWB, and IMU Technologies"

_sensors, 2023, doi:10.3390/s23208598_

Round 1

Reviewer 1 Report

The manuscript is focused on Indoor Positioning Technologies. In particular, four different technologies (Wi-FI, UWB, BLE, and IMU) have been analyzed, In addition, Data Fusion and Machine Learning techniques have been considered to try to overcome some limits of the traditional technologies. The paper is interesting and well structured but it could be improved before to be published in the journal. I'd like to provide to co-authors some comments/suggestions:

1) Paper title is too short and synthetic, I suggest to improve it

2) It's important to add details about the literature review done specifying DBs, queries, etc..

3) the subsection Methodology should be improved reporting specific questions that will be answered and commented with the study

4) I suggest to add the technology BLE Mesh

5) check the reference about TPU and BLE5.3.

6)   I suggest to add a new section before "Future Research Directions" to report a synthetic comparison among all considered technologies and solutions trying to answer to questions reported in the initial part of the paper. I suggest to define a short list of parameters and to make a single table that analyze a comparison among all technologies/solutions respect the defined parameters.

The english is good. There are only some typos. 

Author Response

We have tried to address all the questions and a point-to-point response is provided.

Reviewer 2 Report

This paper reviewed four major technologies for implementing an indoor localization system are reviewed: Wi-Fi, UWB, BLE, and IMU. It is not comprehensively studied indoor technologies and has several weaknesses as follows:

- There are many technologies such as RFID, FM radio, Zigbee, VLC, Infrared (IR) and Ultrasound, LoRa, Mobile networks (4G, 5G), Sensor Fusion, Acoustic-Based, etc. that were not covered.

-In Related works: You may also add these works: "A three-dimensional pattern recognition localization system based on a Bayesian graphical model" and "An overview of indoor localization technologies: Toward IoT navigation services"

-Points 2 and 3 are not considered significant contributions in subsection 1.2.

Section 2 is good but still needs to include other technologies that are above mentioned. Also, The tables and figures should be well-placed according to their subsections. It is very difficult to follow. In addition, you need to include several papers that were already published in 2023 to keep this paper up to date. 

Section 3 is very short and should be combined with Section 2. Also, you should summarize which technology is better in terms of Accuracy, Robustness, Scalability, Security, and Complexity. 

Section 4. Data Fusion is not well organized and written and it needs more improvements. Data fusion is the process of integrating and combining data from multiple sources (can be the same technology with multiple sources OR different technologies with multiple sources) to obtain a more accurate, complete, and reliable representation of a situation. For example WiFi and FM radio, WiFi and CSI, Wifi and 5G, BLE and UWB, RFID and sensor fusion and many other hybrid technologies. The authors only discussed 4.1. Crowdsourcing.

Section 5. Machine Learning is not well discussed and needs also more improvements. The authors require to show how can ML improve accuracy using various indoor localization technologies. Figure 3 is only showing ML with BLE. I suggest forming a table that can summarize all discussed technologies with ML.

The reference is missing in Processing Unit (TPU)[? ].

Section 6. Future Research Directions are not discussed in future directions. it seems to discuss the challenges. Future Research Directions are such as 5G and Beyond Integration, Massive-Sensor Fusion, Privacy and Security/ML, Augmented Reality (AR) Integration, Power-Efficient Solutions and Context-Aware Localization, etc.

Author Response

(The authors gave the same response as above.)

Reviewer 3 Report

The authors' research is active and focused on an important area. But the article contains significant shortcomings.

1) The introduction of Table 1 is far from the table. Also, this is not enough. Table 1 should be explained in more detail.

2) This article is a survey study. At the end of the introduction, previous similar surveys or reviews should be mentioned. Differences from these previous studies should be highlighted.

3) Section 1.1 is very inadequate. It should be one of the most important parts of this article. In addition, visual-based indoor localization is also a very active area. Why was the visual work not mentioned? VO, VIO, VSLAM, or VISLAM studies are also very important for indoor localization. Therefore, recent studies in these areas should also be considered. For example, the works of TransFusionOdom, HVIONet, StereoVO, VIIONet should be mentioned.

4) In the introduction, SLAM and odometry applications should be mentioned. For this, the following study can be used: https://doi.org/10.1007/978-3-030-75472-3_7. In addition, the contribution of artificial intelligence to indoor localization should be added by supporting references.

5) The ML mentioned in the summary is important for indoor localization. But in recent studies, deep learning is more intense. Either deep learning should be included, or the more general expression artificial intelligence should be used.

6) A discussion section is required that includes trends, deficiencies, a general comment on different methods and different sensors, etc. in the current studies.

7) Figure 1 should also include the number of documents of camera-based studies including visual applications.

Author Response

(The authors gave the same response as above.)

Round 2

Reviewer 1 Report

The manuscript was revised and improved taking into account the previous comments.

Author Response

Thank you for taking the time to read our paper and providing positive feedback.

Reviewer 2 Report

R1: There are no valid reasons why you choose only four indoor technologies. There is no common term. Moreover, the title is not clear " On indoor localization using.....". The title should be changed.  suggest to include other technologies if there are no strong valid reasons for choosing these four technologies. 

R2: There is a serious mistake in reference [25], you wrongly wrote the authors' names. 

R4: You have not included the suggested technologies that I mentioned: such as RFID, FM radio, Zigbee, VLC, Infrared (IR) and Ultrasound, LoRa, and Mobile networks (4G, 5G).

R5:  Summarizing technologies is better in terms of Accuracy, Robustness, Scalability, Security, and Complexity. Please refer to this paper: A review on wireless emerging IoT indoor localization.

R6: This comment is not well addressed as I mentioned" Data Fusion is not well organized and written and it needs more improvements. Data fusion is the process of integrating and combining data from multiple sources (can be the same technology with multiple sources OR different technologies with multiple sources) to obtain a more accurate, complete, and reliable representation of a situation. For example WiFi and FM radio, WiFi and CSI, Wifi and 5G, BLE and UWB, RFID and sensor fusion and many other hybrid technologies."

R10: Future directions should be expanded and also add more directions. 

Author Response

Please check the attached file. Hopefully, we have addressed all the comments.

Reviewer 3 Report

The authors successfully revised the article. However, the following comment should have been made more carefully. After this revision is made successfully, the article can be accepted.

“Section 1.1 is very inadequate. It should be one of the most important parts of this article. In addition, visual-based indoor localization is also a very active area. Why was the visual work not mentioned? VO, VIO, VSLAM, or VISLAM studies are also very important for indoor localization. Therefore, recent studies in these areas should also be considered. For example, the works of TransFusionOdom, HVIONet, StereoVO, VIIONet should be mentioned.” 

Author Response

Please find the attached file. I hope we have addressed all your comments. 
